

# Analysis of slope processes in the Vallcebre landslide (Eastern Pyrenees, Spain) by means of Cross Correlation Function applied to high frequency monitoring data

Marco Mulas[1], Jordi Corominas[2], Alessandro Corsini[1], and Jose Moya[2]

[1]Dept. of Chemical and Geological Sciences, University of Modena e Reggio Emilia, UniMoRe, Modena, 41125, Italy
[2]Division of Geotechnical Engineering and Geosciences, Technical University of Catalonia-BarcelonaTech, Barcelona, 08034, Spain

*Correspondence to:* Marco Mulas (marco.mulas@unimore.it)

**Abstract.** The Vallcebre landslide is a large slow moving translational slide in the Eastern Pyrenees (Spain). In this work, the Cross-Correlation Function (CCF) method was used in order to quantitatively investigate the time-lagged correlation between high frequency monitoring data on rainfall, piezometric and displacement with the objective to evidence hydro-mechanical processes occurring along the slope. The CCF is a signal processing tool for measuring similarities of time-series waveforms as function of an applied time-lag. Specifically, it was applied in Vallcebre landslide to a 3 years long time series of monitoring data, from 1999 to 2001, with a sampling frequency of 20 minutes. Data were measured in three boreholes instrumented with automated wire-extensometers and piezometers and a rain gauge. The boreholes are lined up down the slope and following the displacement direction, which allowed investigating transfer of landslide mass and groundwater along the slide. Several combinations of time series were analysed: rainfall vs. displacement; rainfall vs. piezometric depth; piezometric depth vs. displacement. Moreover, correlation analysis of displacement and piezometric depth between boreholes was also performed. The CCF analysis highlighted and constrained in time a dual triggering mechanism in which factors controlling movement change along the slide: movement in the lower landslide zone is predominantly influenced by toe erosion whereas in the intermediate and upper landslide zone movement is mostly controlled by groundwater recharge and flow.

## 1 Introduction

High-frequency monitoring data regarding displacement and groundwater levels in an active landslide can be crucial for risk management and for early warning (Damiano et al. 2012; Stähli et al., 2015). At the same time, they can support the comprehension of slope dynamics by providing benchmark values for the calibration of physical based models (Van Asch et al., 2009; Eichenberger et al., 2013; Abellan et al., 2015) or by allowing a statistical analysis of cause-effect relationships (Segoni et al., 2014; Manconi and Giordan, 2015; Corsini and Mulas, submitted).

The Cross Correlation Function (CCF) is a statistical tool that allows the time-lagged relationships between time-series of monitoring data to be to assessed. It can be adopted in a wide range of applications (Costa et al. 1992, Vampola 1998, Vorburger et al. 2011, Hyde et al. 2012, Zhang et al. 2015). The CCF has been applied to landslides for measuring debris flow



velocity based on various monitoring devices (Arattano and Marchi, 2005), for the analysis of time response of a landslide to meteorological events (Lollino et al. 2006), for correlating meteorological and rock fall databases (Delonca et al., 2014), for the evaluation of meteorological triggers of large landslides in glaciomarine clay (Gauthier and Hutchinson, 2012). In this work, the CCF is used in order to quantitatively investigate the time-lagged correlation between rainfall, piezometric level and

5 displacement at the Vallcebre translational slide (Spain), so to highlight key factors controlling landslide activity in different points of the slope. High frequency time-series of monitoring data in the period January 1999 to January 2001 have been used. This paper illustrates in detail the results obtained and discusses their significance on a slope stability processes perspective.

## 2   Vallcebre case study

The Vallcebre landslide is located $140$ km north of Barcelona, in the Eastern Pyrenees (Spain), see figure 1a. A detailed
characterization of the landslide is given in Corominas at al. (2000, 2005). It is a slow-moving translational landslide involving a total area of about $0.8$ km$^2$ between $1250$ m and $950$ m elevation. Displacement monitoring in the period 1995 to 2000 (D-GPS surveys and down-hole wire extensometers) provided the first evidences that movement rates are progressively higher from the upper to the lower part of the landslide (Gili et al., 2000). This fact, coupled to the presence of secondary scarps and graben-like zones, allowed Corominas et al. (2005) to separate the landslide in three main units: upper, intermediate and
lower (Fig. 1b). The affected bedrock is clayey siltstone with veins of gypsum overlaying limestone. Boreholes drilled in the landslide area have indicated that the sliding surface is located at around $20$ m depth inside a $1$ to $6$ m thick horizon of fissured clay-shales over which siltstone and gypsum lenses up to some tens of meters wide can also be found (Corominas et al., 2000). On the other hand, borehole cores also indicated that the landslide body is mostly constituted by the clayey siltstones covered by colluvial debris (Corominas et al., 2005). The piezometric depth, measured with reference to the ground level, shows a quite
variable baseline value along the slope: around $4.1$ m depth in S9, $5.9$ m depth in S2 and from $4$ to $3$ m depth in S4 (Fig.2b). A qualitative correspondence of peak increases to rainfall events can be observed (Fig.2b). They generally occur around periods of high rainfall, but not all rainfall events determine relevant groundwater changes and the magnitude of peaks differ between boreholes; i.e. higher in S2 and S9 and lower in S4. This latter borehole is located next to the extension zone, consisting of a graben and a crack system, developed at the bottom of the main scarp of this landslide unit (Fig 1). Low-magnitude response in
S4 is probably due to the high permeability in the graben zone, which was confirmed by pumping tests (Corominas et al. 2008). In addition, this zone has a topographic gradient towards the SW side of the slide, which allows a lateral drainage out of the slide (Corominas et al., 2005). Higher permeability here favours rapid infiltration of rainfall into the landslide but, also, it may drain out more easily (Corominas et al. 2008). It is worth noting that the response of piezometers to rainfall is affected by the bias in time due to the piezometer type, its geometry and the permeability of the geologic media around it. Empirical formulas exist
in order to determine the time-lag necessary to reach inside the piezometer the $90\%$ of the occurred piezometric level variation (so called $t_{90}$) (Hvorselv, 1951; Butler and Mathias, 2006). The assessment of $t_{90}$ is essential in order to correctly interpret the time-lagged cross correlation of piezometric level between boreholes and with rainfall and displacement. Estimates of $t_{90}$ in the Vallcebre piezometers have considered the transversal area of the well-screen (A, based on the open pipe piezometer



radius R), the shape factor of the piezometer (F, calculated according to Butler and Mathias, (2006)) and the local permeability of the soil around the piezometer (K), see figure 3. The known piezometer characteristics and the hydrologic data presented by Corominas at al. (2008) have been considered to such purpose. Specifically, the following assumptions have been made in order to define the permeability of the soil around the piezometer and, consequently, assess $t_{90}$ of each piezometer:

- Piezometer S4 (open standpipe 11 m long, diameter $5 \times 10^{-2}$ m, fissured from $-1$ to $-11$ m): it is located in the area that Corominas et al. (2008) identify as "graben" zone in their hydrologic model. The authors provide a $K = 2 \times 10^{-6}$ m s$^{-1}$ from pumping tests and conclude that the hydrological model reaches the best fit to monitored data (case 2) if a $K = 1.2 \times 10^{-6}$ m s$^{-1}$ is used in this zone. Therefore, $K = 1.2 \times 10^{-6}$ m s$^{-1}$ has been used to calculate a $t_{90}$ of $8.0$ hours (fig. 3d).

- Piezometer S2 (open standpipe 15 m long, diameter $5 \times 10^{-2}$ m, fissured from $-1$ to $-15$ m of depth): it is located in the area that Corominas et al. (2008) identify as "domain" zone with over imposed a superficial "piping" zone related to lenses of gypsum up to 5 m thick. The authors provide a $K = 5.7 \times 10^{-8}$ m s$^{-1}$ from pumping tests that should be considered representative of the landslide "domain" zone. The authors conclude that the hydrological model provides the best fit to monitored data (case 2) if a $K = 8.1 \times 10^{-8}$ m s$^{-1}$ is assigned to the "domain" zone and a $K = 4.3 \times 10^{-6}$
m s$^{-1}$ is assigned to the "piping" zone. Therefore, we considered that the piezometer intersects the "piping" zone in the range -1 ÷ -3 m below the ground and the "domain" zone in the range -3 ÷ -15 m below the ground. According to this, an equivalent horizontal permeability (accounting proportionally for both domains) of $6.8 \times 10^{-7}$ m s$^{-1}$ has been used to calculate $t_{90}$ of 10.6 hours (fig. 3d).

- Piezometer S9 (open standpipe 15 m long, diameter $5 \times 10^{-2}$ m, fissured from $-1$ to $-15$ m): it is located in the area
that Corominas et al (2008) identify as "domain" zone with over imposed a superficial "piping" zone related to lenses of gypsum up to 5 m thick. The authors do not provide results of pumping tests. On a geomorphic basis, S9 is located toward the toe of the landslide, where compression should further reduce permeability. However, Ferrari et al. (2011) stress the key-role of the toe erosion by the Vallcebre stream for the triggering of the landslide. This means that this area is subject to alternated phases of compression and extensions that, in the long term, might result in permeability values
not different from these of S2. Therefore, the same considerations and assessments made for S2 have been used in order to estimate an equivalent horizontal permeability of $6.8 \times 10^{-7}$ m s$^{-1}$ and a $t_{90}$ of 10.6 hours (fig. 3d).

It should be finally stressed that if S2 and S9 would be considered surrounded only by low permeability materials of the "domain" zone ($8.1 \times 10^{-8}$ m s$^{-1}$) the resulting $t_{90}$ would be in the order of more than 80 hours; this estimate is believed unrealistic, as the cross-correlation between rainfall and piezometric level indicates time lags not longer than 22 hour.

## 3 Dataset

The monitoring dataset analysed in this paper refers to the lowest of the landslide units identified by Corominas et al. (2005). It derives from a rain-gauge installed at the landslide site and from three boreholes, each equipped with an in-hole automated





wire-extensometer and an open standpipe piezometer with a pressure transducer sensor (identified as S2, S4 and S9, see Fig. 1c). The time-series of monitoring data cover the period January 1999 to January 2001 with a sampling interval of 20 minutes. In these three years of monitoring, wire extensometer S9, close to the toe of the landslide, reached 0.725 m of displacement; S2, in the intermediate part of the unit, reached 0.658 m and S4, close to the head scarp of the unit, reached 0.638 m. All

the wire extensometers display a seasonal trend, with accelerations in spring and fall (periods with higher rainfall rate) and a number of short term acceleration periods after specific precipitation events (Fig.2a).

## 4   Methods

The Cross Correlation is a widely applied technique in signal processing for quantitatively measure the similarity between two times-series as a function of the time lag of one time-series respect the other. An exhaustive demonstration of CCF usage for

discrete time-series analysis is presented in Shumway and Stoffer (2011). In practice, given two discrete time-series X(t) and Y(t), the CCF analysis generates a plot of the correlation value between the time-series as a function of the lag (correlogram) allowing to assess the relationship between the two time-series on the basis of:

1. the correlation value in a scale from $-1$ to $1$ (where $0$ indicates no correlation, positive values indicate direct correlation with mutual increase of both time-series as in Fig.4b, and a negative value reflects inverse correlation, with the increase

of one time-series correlating to the decrease of the other, as in Fig. 4a);.

2. the time-lag "h" at which the cross correlation reaches its maximum (time-lag at max CCF). The time-lag can be positive or negative. This determines which time-series is leading the other in the time domain. For instance, if the time-lag between $X(t)$ and $Y(t+h)$ is positive, it means that $X(t)$ is antecedent to $Y(t)$ (X "leads" Y, see in Fig. 3.4a). Negative time-lags indicate that $X(t)$ is subsequent to $Y(t+h)$ (X "lags" Y, see in Fig. 4b);

3. the statistical significance threshold, i.e. the CCF 95% values above and below which the cross-correlation is significant. The threshold is computed on the basis of the statistical population considered, i.e. the number of records of the time-series;

4. the time-lags corresponding to the 5% interval around the maximum CCF value (i.e. time lags at max CCF 5%, that are the red dashed vertical lines in Fig. 3.4a and 3.4b). This parameter was calculated in this work in order to assess the

robustness of the leading time series assessment based on max CCF time-lag. For instance, if the time-lags at 5% from max CCF maintain the same sign (positive or negative), it indicates that that the leading time series assessment does not change even if a 5% variability of CCF around its maximum is considered.

The preliminary processing of the Vallcebre landslide dataset included the application of a Fourier analysis of the time series in order to identify and filter possible recurrent noises by still preserving the original time-sampling interval of the dataset. A

30 h low-pass filter was applied to remove from time series a 24 h frequency noise (Fig. 5). The noise is believed to be related with electrical disturbances in the dataloggers induced by the daily thermal oscillation. Subsequently, synchronous subsets of



every time-series (Cumulated rainfall, Piezometric depth and Displacements), for all monitoring points were created by setting the starting time at $00:00$ of the $01/01/1999$ and the ending time at $00:00$ of the $01/01/2002$ keeping the original time sampling of 20 minutes. Such operations, as well as the CCF analysis were performed with the aid of scripts in R environment (R Development Core Team, 2015).

In order to highlight the dynamics occurring within the Vallcebre landslide, the CCF analysis took into consideration the following different combinations of time series:

- rainfall vs. displacement (at S2, S4 and S9, see Fig.6a);

- rainfall vs. piezometric depth (at S2, S4 and S9, see Fig. 6b);

- piezometric depth vs. displacement (at S2, S4 and S9, see Fig. 6c);

- displacement vs. displacement (S4 vs. S2, S4 vs. S9 and S2 vs. S9, see Fig. 6d);

- piezometric depth vs. piezometric depth (S4 vs. S2, S4 vs. S9 and S2 vs. S9, see Fig. 6d);

For all these combinations, the value of CCF, the time-lag at max CCF, the time-lags at max CCF $5\%$ and the statistical significance limits were obtained and analysed.

## 5   Results

Tables 1 to 5 summarise the key results obtained by mean of CCF of the several time series combinations. It might be noticed that in some time series combinations the maximum CCF values are quite low (less than $1 \times 10^{-1}$) whereas in other are rather high (up to $0.8$). Nonetheless, the results obtained with low CCF values were still considered significant since the correlograms show patterns where the identification of a maximum is still rather clear and, at the same time, the max-correlation absolute values are of at least one order of magnitude above the statistical significance thresholds.

**5.1   Rainfall ($X_t$) vs. Piezometric depth ($Y_{t+h}$)**

Results are presented in Table 2. The max CCF values are quite low, but still they are significant as they are at least one order of magnitude above the statistical significance threshold. The time-lags are always positive, both at max CCF and at max CCF $5\%$, indicating that rainfall time series are quite certainly leading the displacement time series in the landslide. The time-lag at max CCF in S4, at the top of the considered landslide unit, is $+26$ hours. Moving downslope, the time-lags at max CCF

decrease to $+17$ hours in S2 and $+15$ hours in S9, at the bottom of the landslide unit.

**5.2   Rainfall ($X_t$) vs Displacement ($Y_{t+h}$)**

Results are presented in Table 3. The max CCF values are negative, since a decrease of piezometric depth from the ground corresponds to an increase of displacement. The max CCF values are rather high in S2 and S9 ($0.82$ and $0.70$ respectively)

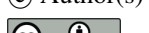



while it is much lower in S4. Nevertheless, even in S4, the CCF value is one order of magnitude above the statistical significance threshold. The time-lags at max CCF are all positive, indicating that the piezometric depth is leading displacement. The time-lag at max CCF in S4 is +15 hours, 17.6 hours in S2 and 9.6 hours in S9. Nevertheless, the time lags at max CCF. 5% range from positive to negative values in S2 and S9, indicating that the assessment of the leading time series is affected by some

degree of uncertainty. The low CCF value obtained in S4 can be explained by the high permeability of the ground around the pipe. As it was mentioned above, extension derived features (a graben and open cracks) are found in this part of the slide, close to S4 borehole, and open fissures are probably also present around it. High permeability determines a quite limited increase of the piezometric level following a rainfall event as water is rapidly drained. On the other hand, in boreholes S2 and S9, far from the graben area, the lower permeability gives a much pronounced response (i.e. increase) of the piezometric level and

consequently a better CCF value respect to displacements.

### 5.3 Piezometric depth ($X_t$) vs. Displacement ($Y_{t+h}$)

Results are presented in Table 4. The max CCF values are high (around $0.8$ in every combination) and therefore two order of magnitude above the statistical significance threshold. The time-lags at max CCF are always negative, indicating that displacement of the wire extensometer located downslope is leading the displacement of the wire extensometer located upslope.

The time-lag at max CCF between S4 and S2 is quite small, less than one hour, indicating that displacement time series in S4 and S2 show peaks that are almost synchronous; this can result, at least in part, from the small distance existing between these two boreholes (50 m down the slope). On the other hand, the time-lags at max CCF between S4 and S9 (separated 150 m) and S2 and S9 (separated 100 m) are significant: $-17$ and $-13$ hours respectively. This, in other terms, seems to indicate a retrogressive evolution of movements starting from the bottom of the slope. The time-lags at max CCF 5% range from positive

to negative values, indicating that the assessment of the leading time series is affected some degree of uncertainty. Nevertheless, the ranges between lower and upper boundary of time-lags at max CCF 5% are predominantly in the negative values, thus it is reasonable to consider quite reliable the assessment of retrogressive evolution with time series of displacement in downslope boreholes leading time series of displacement in upslope boreholes.

### 5.4 Displacement ($X_t$) vs. Displacement ($Y_{t+h}$)

Results are presented in Table 5. The max CCF values are low (less than $0.1$) except in the case of S2 vs S9, where it is quite high ($0.7$). For instance, in the case of S4, the low-magnitude response can be related to the fact that is located in the graben zone. Therefore, rainfall may rapidly percolate into the landslide but also it may drain more easily out towards a lateral boundary of the slide. As a consequence, the S4-S2 and S4-S9 CCF low results can be influenced by the low magnitude of response of S4 to piezometric variations. Nevertheless, in every combination, the CCF values are at least one order of magnitude

above the statistical significance threshold. The time-lags at max CCF value are always positive, indicating that peaks of the piezometer located upslope lead peaks of the piezometer located downslope. The time-lag at max CCF value between S4 and S2 is equal to 3.9 hours. On the other hand, the time lags at max CCF value between S4 and S9 and S2 and S9 are significant, $+7.9$ and $+4$ hours respectively. This, in other terms, could be interpreted as the effect of a downslope directed pore pressure



transmission. However, the time-lags at max CCF 5% range from positive to negative values in every combination, indicating that the assessment of the leading time series is somehow uncertain.

## 5.5 Piezometric depth ($X_t$) vs. Piezometric depth ($Y_{t+h}$)

Results are presented in Table 5. The max CCF values are low (less than $0.1$) except in the case of S2 vs S9, where it is quite high ($0.7$). For instance, in the case of S4, the low-magnitude response can be related to the fact that is located in the graben zone. Therefore, rainfall may rapidly percolate into the landslide but also it may drain more easily out towards a lateral boundary of the slide. As a consequence, the S4-S2 and S4-S9 CCF low results can be influenced by the low magnitude of response of S4 to piezometric variations. Nevertheless, in every combination, the CCF values are at least one order of magnitude above the statistical significance threshold. The time-lags at max CCF value are always positive, indicating that peaks of the piezometer located upslope lead peaks of the piezometer located downslope. The time-lag at max CCF value between S4 and S2 is equal to 3.9 hours. On the other hand, the time lags at max CCF value between S4 and S9 and S2 and S9 are significant, $+7.9$ and $+4$ hours respectively. This, in other terms, could be interpreted as the effect of a downslope directed pore pressure transmission. However, the time-lags at max CCF ±5% range from positive to negative values in every combination, indicating that the assessment of the leading time series is somehow uncertain.

## 6 Discussion

As reasonably expected, the cross-correlation analysis has pointed out that rainfall acts as the leading time series for both piezometric depth and displacement (see Fig. 7a and Fig. 7b for a synoptic plot). However, the results obtained by means of CCF analysis can be discussed on the perspective of hydro-mechanical slope processes that are somehow more complicated that a direct relationship between rainfall, groundwater and movements. The presence of lagged response in open-pipe piezometers like those installed in the Vallcebre landslide should be beard in mind during the discussion of the results. Depending on the piezometer type, geometry and on the permeability of the local soil around it, different time lags are necessary to measure the 90% of the occurred piezometric level variation. Nevertheless, it should be taken in consideration that a complete level variation is not necessary for the CCF analysis to determine a time-lag value. On a hydrogeological perspective, it should be noticed that piezometric depth shows the quickest response to rainfall (shorter time-lag) at the top of the landslide unit (S4) while the time-lag increases downslope (S2 and S9) as a function of site permeability (lower at sites S9 and S2 than S4). On the other hand, the time-lags between different piezometers from upslope to downslope range predominantly in the positive values (Fig. 7f). Combining these two results, it is therefore reasonable to consider as a working hypothesis the presence of a downslope directed pore-pressure transfer-wave that, even in substantial absence of water transfer (given the low permeability of landslide materials), determines the major peaks of groundwater levels after rainfall. Therefore, pore-pressure transfer results to be the hydrological key-factor determining groundwater level variations in this portion of the landslide body rather than an actual groundwater filtration involving mass transfer. A tentative estimation of the apparent pore-pressure velocity transfer can be done by considering the time-lags obtained in the cross-correlation between piezometric depths and



the distance between piezometers. On a such basis, the apparent pore-pressure velocity transfer can be estimated as: $2.1 \times 10^{-2}$ m $\times$ s$^{-1}$ from S4 to S2; $4.1 \times 10^{-1}$ m $\times$ s$^{-1}$ from S2 to S9; $3.1 \times 10^{-1}$ m $\times$ s$^{-1}$ from S4 to S9. It can be speculated that the higher velocity obtained in the sector from S2 to S9, in the lowest part of the landslide unit, might be related to the fact that it corresponds to a compression zone where, presumably, the pore-pressure transfer through the landslide body is

more effective. On a mechanical perspective, the cross correlation between displacements recorded by the wire-extensometers indicates that it is quite reliable to consider the assessment of a retrogressive evolution of movement propagation, with time series of displacement in downslope extensometers leading time series of displacement in upslope extensometers (see Fig. 7e for a synoptic view). The aforementioned evidence is in agreement with the results of the landslide evolution model proposed by Ferrari et al. (2011), in which the key-role of the toe erosion by the Vallcebre stream for the triggering of the landslide is

stressed. On a hydro-mechanical perspective, the analysis of the relationship between piezometric depth and displacements confirm the leading role of piezometric depth variation since in all three monitoring sites displacement peaks result to occur after ground water peaks (Fig. 7b-c). Considering all the aforementioned findings, it is therefore reasonable to assume that the response of the Vallcebre landslide to precipitation is determined by two mutually dependent mechanisms. The first forcing factor to act on the slope stability is the piezometric depth variation flowing by means of pore-pressure transfer through the

landslide body from upslope to downslope after rainfalls occurs in the area. Destabilising pore pressure conditions are generated once the piezometric depth peak have reached the landslide toe. The advance of the landslide mass increase instability of the landslide front, which combined with the torrent erosion, cause periodic small failures there, delivering slope material to the valley bottom subsequently evacuated by the stream (Fig. 8). This triggers a retrogressive evolution of slope movements that affects all the lower unit of the Vallcebre landslide.

**7  Conclusions**

The application of the cross-correlation analysis allowed the discussion of the triggering mechanisms responsible for variations in the displacement rates at the lower unit of Vallcebre landslide. A dual mechanism triggering movement accelerations has been identified that includes the mutual influence of the rapid increases of piezometric levels (promoted by a rapid pore-pressure wave-transfer through the lower unit of the Vallcebre landslide) that reduce effective stresses and shear strength along the

sliding surface and the toe erosion operated by the Vallcebre stream (a direct proxy for the discharge of rainfall in the catchment) that generates a retrogressive destabilization of the slope. This high-frequency monitoring dataset of the Vallcebre landslide has been ideal for testing the application of CCF tools for the exploration of the information contained within monitoring time-series. The adopted methodology proved successful in highlighting and quantifying the interdependencies between the variables playing a role in slope stability and it is replicable in any active landslide instrumented with automated monitoring

systems in which time-series including several acceleration events have been collected.



## Appendix A

In this section the correlograms obtained are presented. It is noteworthy to recall that time-lags do not take into account the $t_{90}$ response delay that has been presented and estimated in the Dataset paragraph and that has been considered in Tables from 1 to 5 that present the result of the elaborations. The presented correlograms display different time-lag intervals since we have

5   chosen the time window in order to achieve the best visualization of the CCF pattern for identifying absolute maximum points of correlation.

*Acknowledgements.* This research was carried out thanks to the financial support of the International Mobility Programme 2013 of the University of Modena and Reggio Emilia



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

**Table 1.** Rainfall ($X_t$) vs. Piezometric depth ($Y_{t+h}$) CCF results.

| CCF Rainfall VS Piez. depth | CCF | | Time lap (hours)* | | |
|---|---|---|---|---|---|
| | Max value | Max - 5% | At CCF Maximum | At CCF max - 5% | |
| | | | | Lower boundary | Upper boundary |
| S4 | −0.03 | −0.029 | +2.0 | +0.0 | +4.0 |
| S2 | −0.11 | −0.104 | +5.4 | −0.6 | +16.4 |
| S9 | −0.05 | −0.047 | +11.4 | +3.4 | +23.4 |

CCF 95% Significance Threshold ±0.0007            * time-lag corrected by the $t_{90}$




**Table 2.** Rainfall ($X_t$) vs. Displacement ($Y_{t+h}$) CCF results.

| CCF<br>Rainfall VS Displ. | CCF | | Time lap (hours) | | |
|---|---|---|---|---|---|
| | Max value | Max - 5% | At CCF Maximum | At CCF max - 5% | |
| | | | | Lower boundary | Upper boundary |
| S4 | +0.065 | +0.062 | +26 | +19 | +36 |
| S2 | +0.068 | +0.065 | +17 | +12 | +23 |
| S9 | +0.13 | +0.124 | +15 | +10 | +19 |

CCF 95% Significance Threshold ±0.0007

**Table 3.** Piezometric depth ($X_t$) vs. Displacement ($Y_{t+h}$) CCF results.

| CCF<br>Piez. depth VS Displ. | CCF | | Time lap (hours)* | | |
|---|---|---|---|---|---|
| | Max value | Max - 5% | At CCF Maximum | At CCF max - 5% | |
| | | | | Lower boundary | Upper boundary |
| S4 | −0.068 | −0.065 | +15.0 | +5.4 | +29.6 |
| S2 | −0.82 | −0.78 | +17.6 | −2.4 | +39.6 |
| S9 | −0.70 | −0.67 | +9.6 | −20.4 | +33.6 |

CCF 95% Significance Threshold ±0.0007                    * time-lag corrected by the $t_{90}$

**Table 4.** Displacement ($X_t$) vs. Displacement ($Y_{t+h}$) CCF results.

| CCF<br>Displ. VS Displ. | CCF | | Time lap (hours) | | |
|---|---|---|---|---|---|
| | Max value | Max - 5% | At CCF Maximum | At CCF max - 5% | |
| | | | | Lower boundary | Upper boundary |
| $S4_{t_0}$ vs $S2_{t_0+h}$ | +0.81 | +0.77 | −0.7 | −33.0 | +19.0 |
| $S4_{t_0}$ vs $S9_{t_0+h}$ | +0.83 | +0.79 | −17.0 | −40.0 | +4.0 |
| $S2_{t_0}$ vs $S9_{t_0+h}$ | +0.85 | +0.81 | −13.0 | −33.0 | +6.0 |

CCF 95% Significance Threshold ±0.0007


**Table 5.** Piezometric depth ($X_t$) vs. Piezometric depth ($Y_{t+h}$) CCF results.

| CCF | CCF | | Time lap (hours)* | | |
|---|---|---|---|---|---|
| Piez. depth VS Piez. depth | Max value | Max - 5% | At CCF Maximum | At CCF max - 5% | |
| | | | | Lower boundary | Upper boundary |
| $S4_{t_0}$ vs $S2_{t_0+h}$ | +0.078 | +0.074 | +3.9 | −0.4 | +8.6 |
| $S4_{t_0}$ vs $S9_{t_0+h}$ | +0.035 | +0.033 | +7.9 | −0.6 | +16.6 |
| $S2_{t_0}$ vs $S9_{t_0+h}$ | +0.710 | +0.675 | +4.0 | −31.0 | +41.0 |

CCF 95% Significance Threshold ±0.0007                    * time-lag corrected by the $t_{90}$

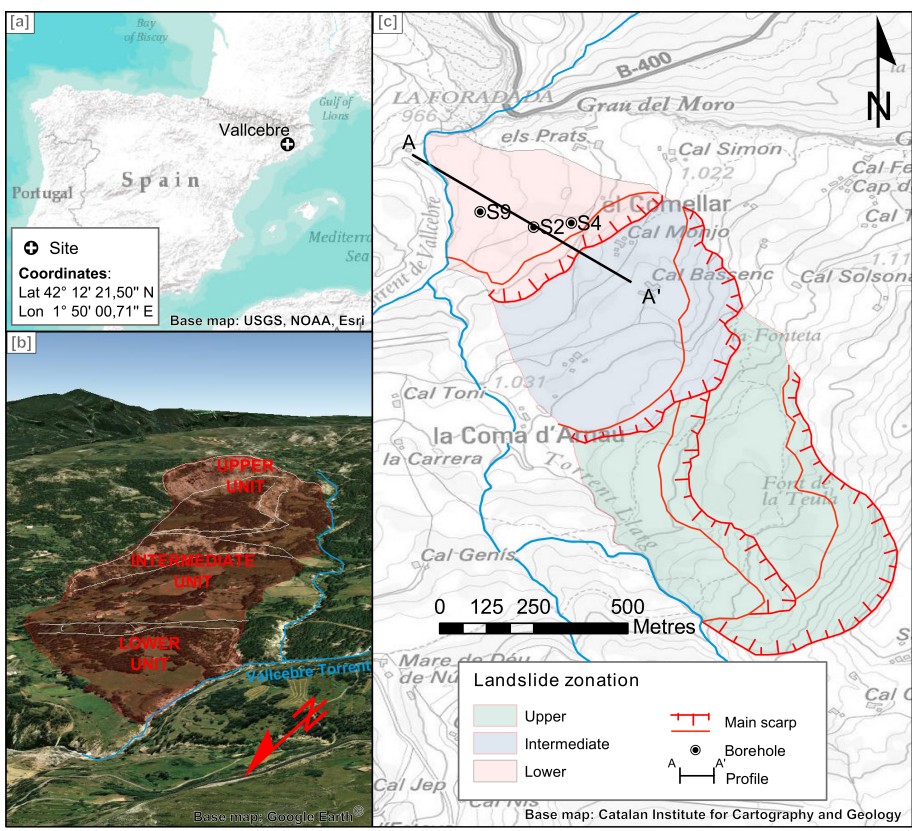

**Figure 1.** ) Location of the Vallcebre Landslide; b) perspective view of the Vallcebre landslide; c) Vallcebre landslide geomorphic zonation with locations of in-situ wire-extensometers/piezometers and geological cross-section investigated in this study (modified after Corominas et al., 2005).




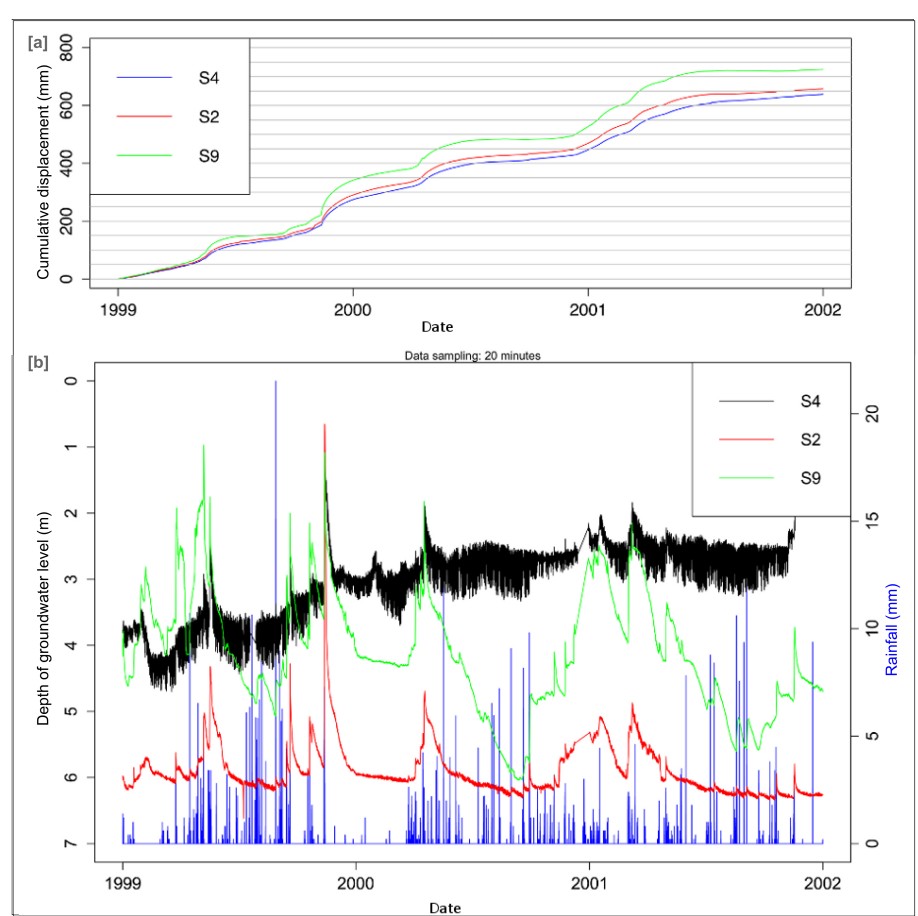

**Figure 2.** a) Cumulative displacements monitored at sites S4, S2 and S9. b) Rainfall recorded at the landslide site versus piezometric depth

variation. All time-series are sampled with a 20 minutes frequency and cover the period from 1/1/1999 to 1/1/2002.





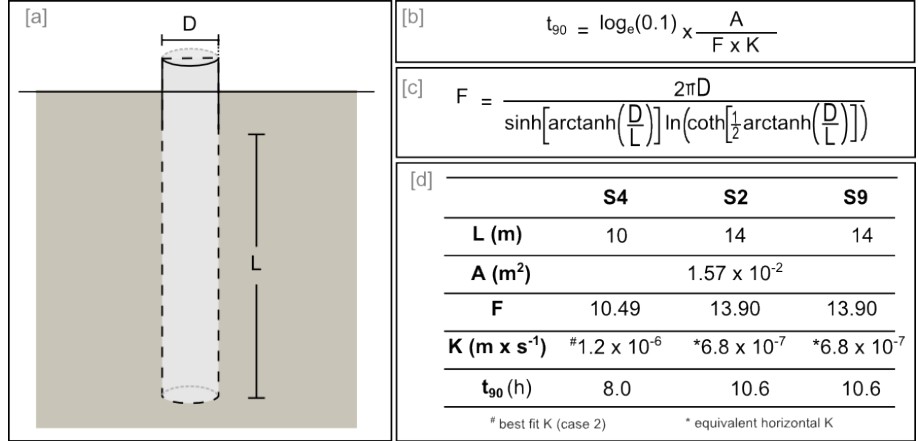

**Figure 3.** a) open pipe piezometer scheme, D and L are the diameter and length of the well-screen (mod. After Hvorslev, 1951); b) formula for the estimation of the t90 considering the transversal area of the well-screen (A), the shape factor of the piezometer (F) and the local permeability of the soil around the piezometer (K); c) shape factor for a well point in uniform soil, refers to scheme of figure 3a; d) geometrical characteristics of well points in the Vallcebre landslide and shape factors (F) plus $t_{90}$ estimated.

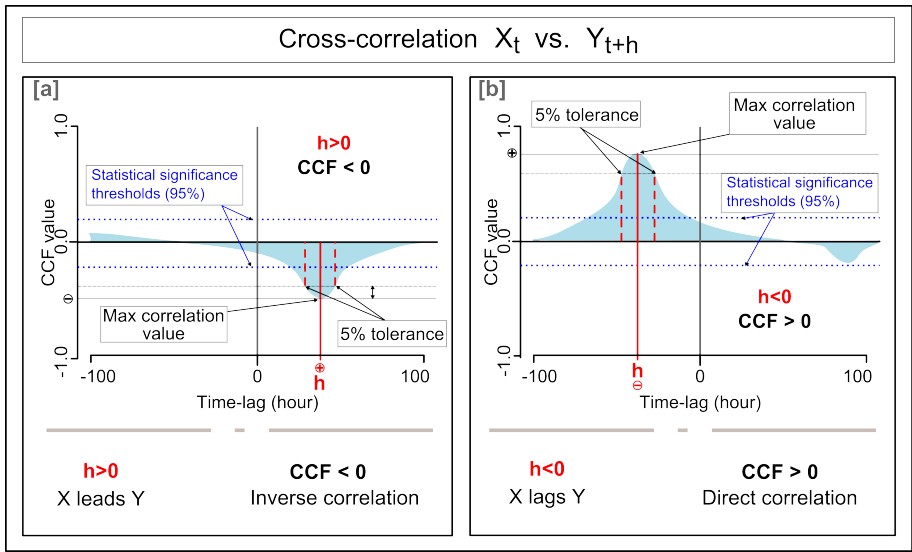

**Figure 4.** Conceptual correlograms illustrating key-features of CCF analysis: "h" is the time-lag corresponding to the max CCF value (red vertical line); 5% CCF tolerance interval and respective time-lag values (vertical red dashed lines); statistical significance interval (horizontal blue dotted lines).





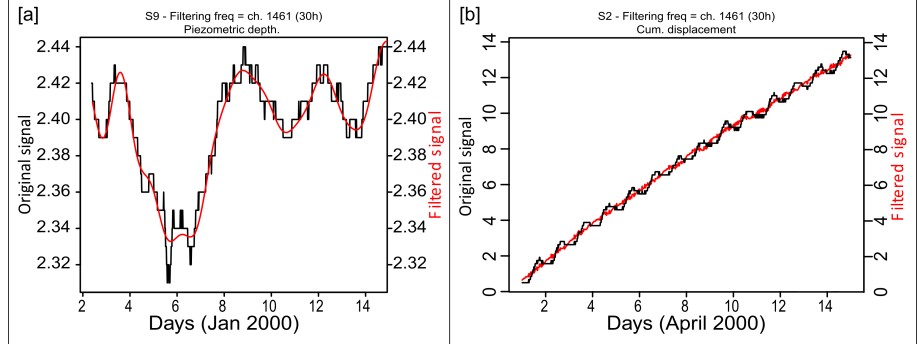

**Figure 5.** Example of Fourier filtering of the Vallcebre data set. Raw data (black lines) show noise at 24 h and higher frequencies. Red lines show data filtered by using a low-pass filter with a 30 h threshold of: a) piezometric depth recorded at site S9; b) cumulative displacement recorded at site S2)

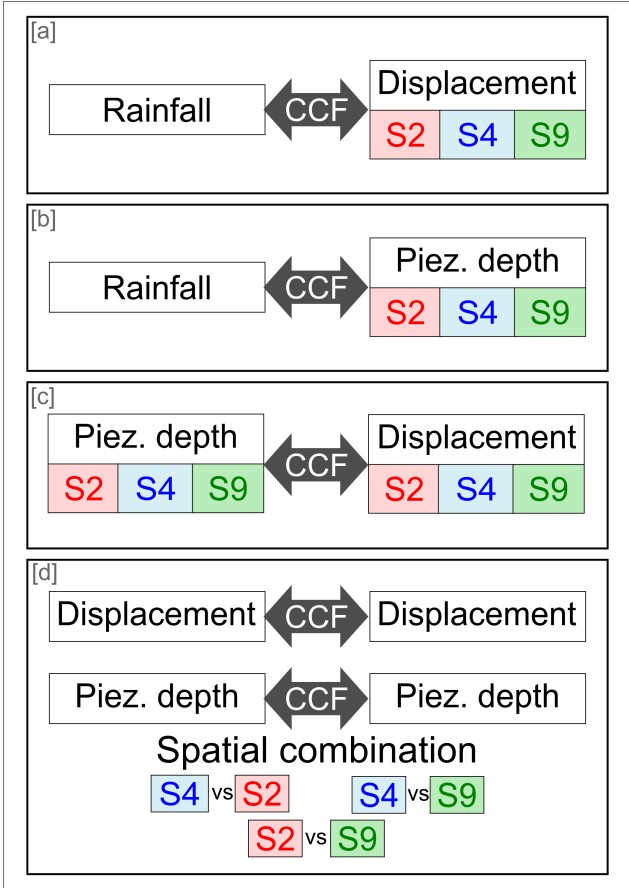

**Figure 6.** Combinations of CCF tasks performed




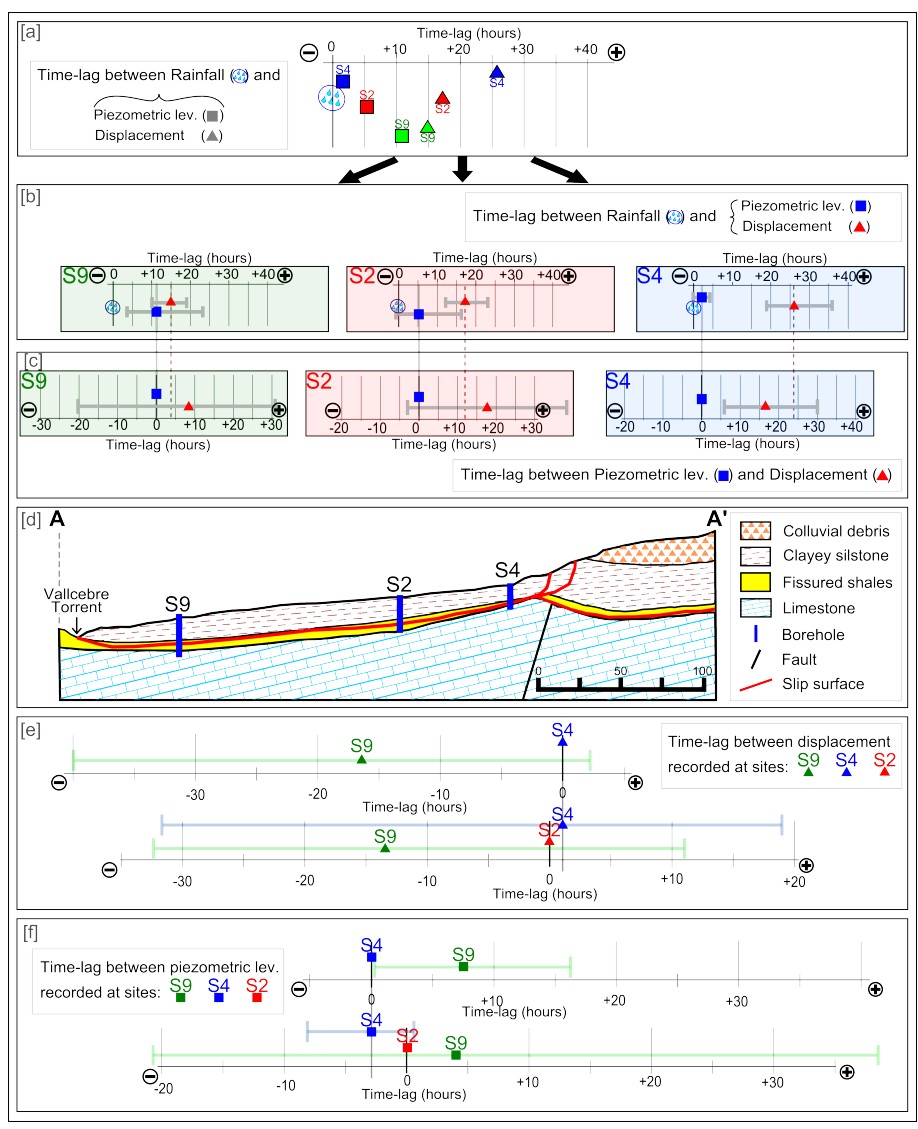

**Figure 7.** Spatio-temporal representation of results of the cross-correlation analysis; a) and b) representation of time-lags between rainfall and Piezometric depth/Displacement; c) identification of the leading time-series between displacement and piezometric depth in the three bore-holes; d) cross-section of the lower unit of the Vallcebre landslide with location of the boreholes (modified after Corominas at al. 2005); e) representation of time-lags between displacement records; f) representation of time-lags between piezometric depth records



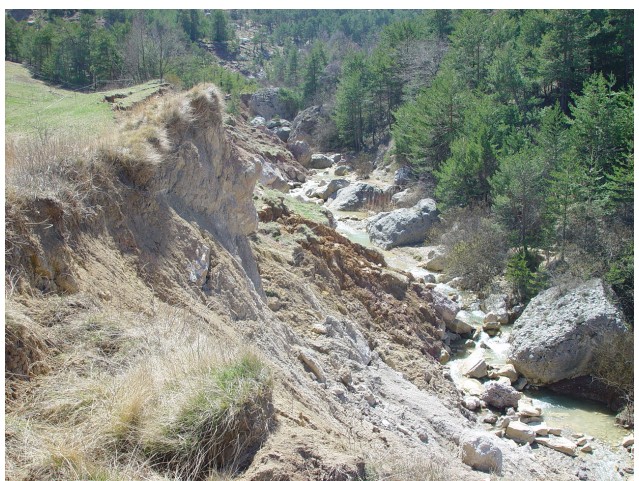

**Figure 8.** Picture of the landslide toe (taken on April 2003)

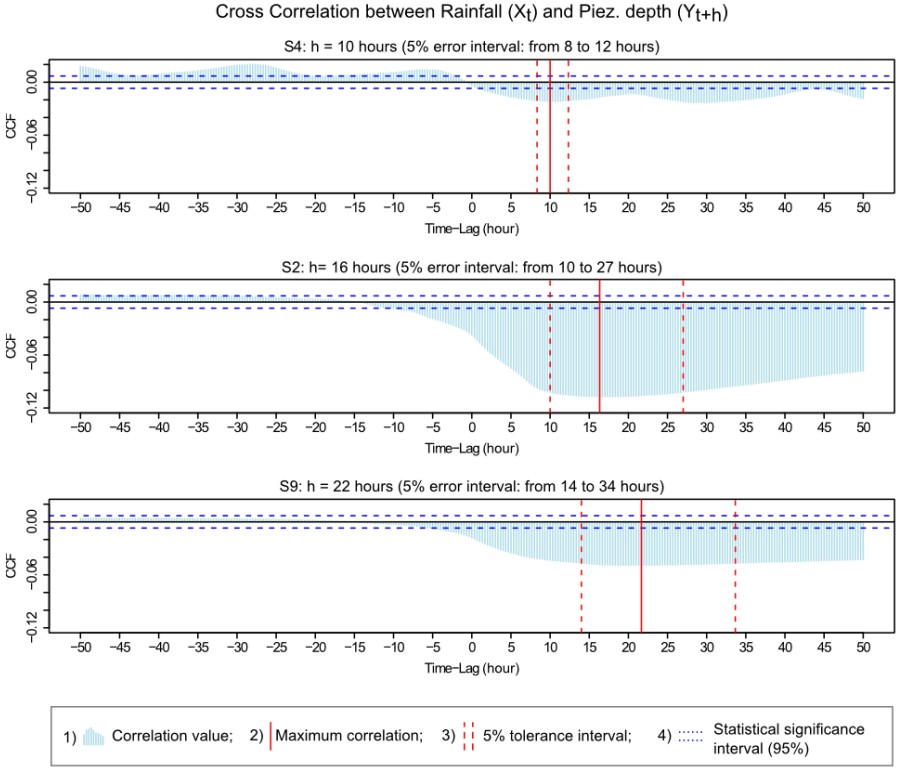

**Figure A1.** Results from the CCF analysis made taking into account rainfall versus piezometric depth recorded at boreholes S4, S2 and S9.





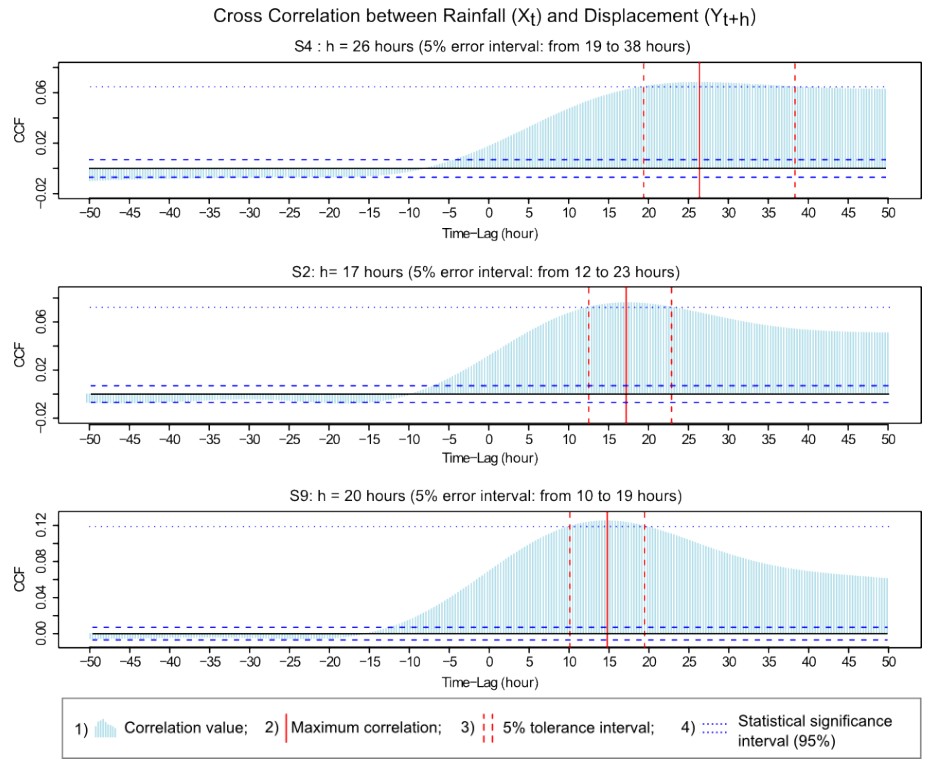

**Figure A2.** Results from the CCF analysis made taking into account rainfall versus displacements recorded at boreholes S4, S2 and S9.





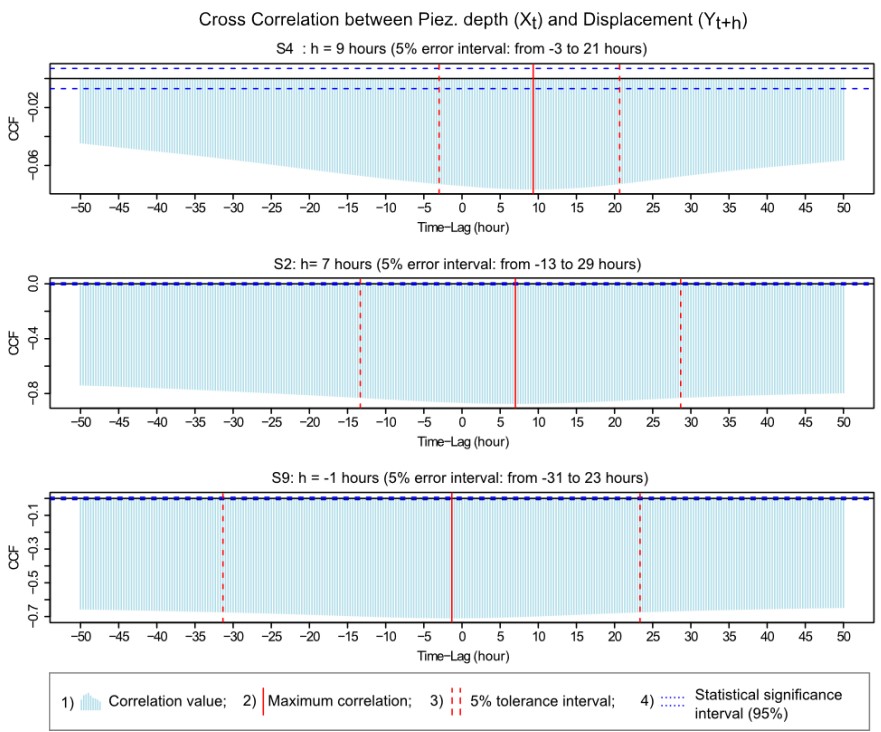

**Figure A3.** Results from the CCF analysis made taking into account piezometric depth versus displacements recorded at boreholes S4, S2 and S9.

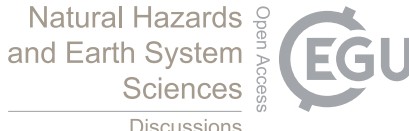



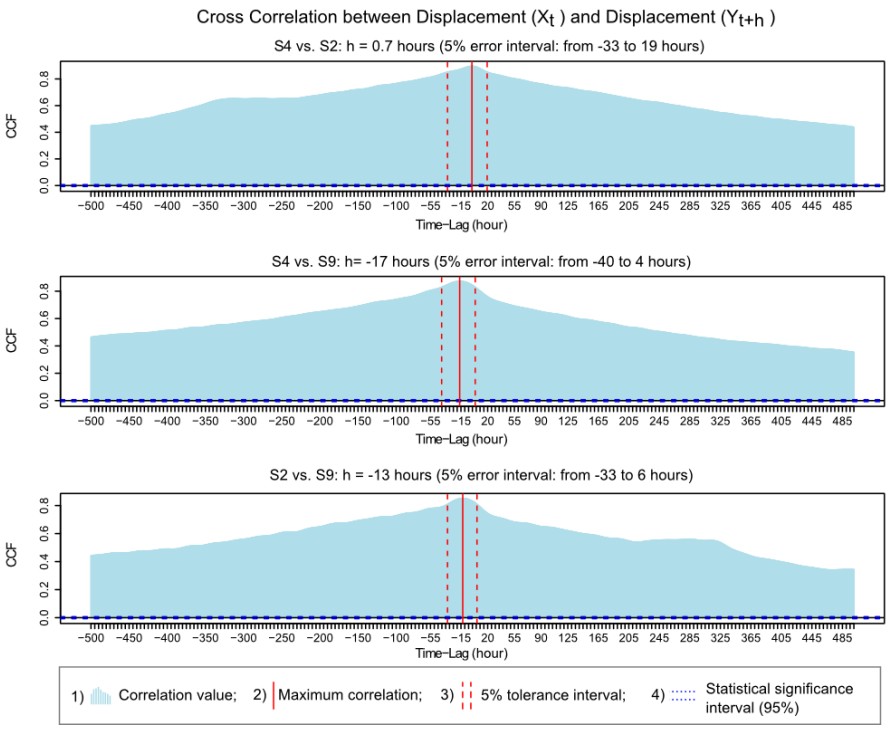

**Figure A4.** Results from the CCF analysis made taking into account displacements recorded at different boreholes, investigating the combinations S4-S2, S4-S9 and S2-S9.



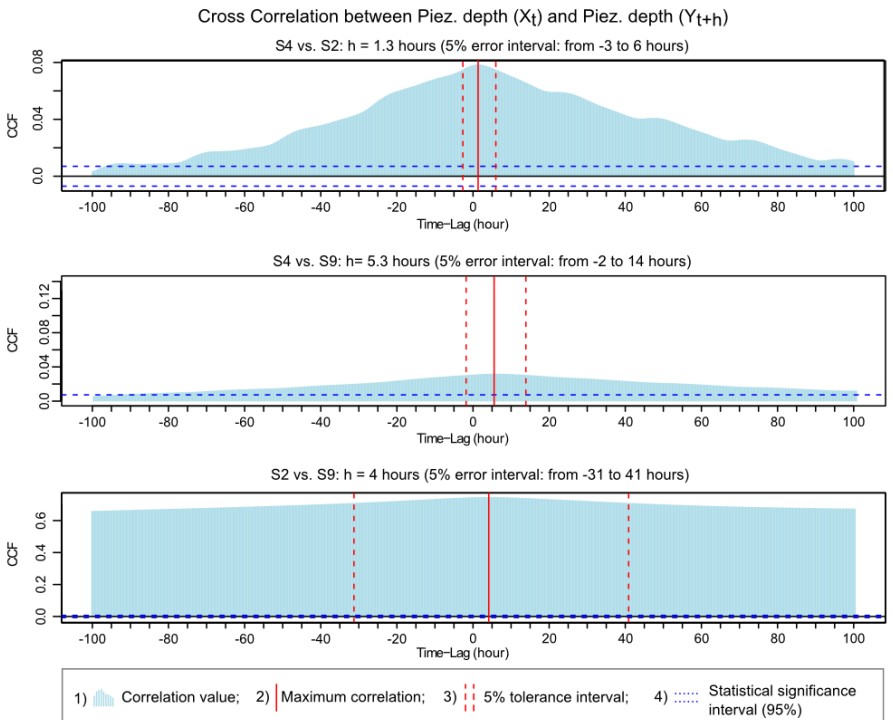

**Figure A5.** Results from the CCF analysis made taking into account piezometric depth recorded at different boreholes, investigating the combinations S4-S2, S4-S9 and S2-S9.