# Peer review of "Analysis of slope processes in the Vallcebre landslide (Eastern Pyrenees, Spain) by means of Cross Correlation Function applied to high frequency monitoring data"

_Natural Hazards and Earth System Sciences, 2016_

## Referee Comment (RC1) · Anonymous Referee #1 · 24 Aug 2016

General comments

The paper deals with the statistical analysis of time series collected in three boreholes of the lower part of a deep-seated slow-moving landslide located in Spain (Vallcebre landslide). The authors investigate the cross-correlation among the series of rainfall, piezometric depth and displacements, in various combinations and comparing measurements of the same variable in the different boreholes. The paper is generally well written, though there is a mistake in the organization of the results section (see details "specific comments"). It certainly fits within the scope of NHESS. The degree

of novelty of the presented material is not so significant, since monitoring of the Vallcebre landslide has been presented in other papers (Corominas et al., 2005) and the cross-correlation technique is standard statistical exploratory tool. The novelty of the paper may be represented by: a) the presentation of data for other periods that those presented in other papers b) the use of cross-correlation function on these series to investigate the interdependence between the measured variables. Nevertheless, one main issue in applying (linear) cross-correlation techniques is the underlying assumption of linear correlation, which is not always valid in the case of the paper. I conclude that the paper may be submitted only after mayor revisions, provided substantial modifications following the specific comments given below.

Specific comments

Introduction: Literature cited in the paper may be enriched (for instance, for the early warning and the modeling part – first paragraph of introduction, P1 L15-19) P1 L20: Cross-correlation is a quite standard statistical tool (see Handbook of hydrology, Salas, 1993 - this must be cited)

P4 L20: Please provide more details on how the confidence lines have been determined (statistical significance threshold), i.e. the formula used

P4 L23: Why a 5% threshold has been chosen, and not another one? The authors should possibly investigate if statistical tests aimed at verifying the significance of cross-correlation variation do exist. Though this issue may significantly change the time-lag interval around the maximum CCF value, the 5% threshold may be however accepted. What I just ask to the authors is to possibly justify this value and to verify the existence of above-mentioned statistical tests

P5 L15-20: "it may be noticed that in some time series combinations the maximum CCF values are quite low". This is probably due to the presence of non-linear (instead of linear) correlation among some of the variables. In fact, if one thinks to the Richards' infiltration equation linking rainfall to piezometric height the relationship between this

pair of variables is non-linear (e.g Iverson, 2000; D'Odorico et al., 2005; Baum et al., 2010; Peres and Cancelliere, 2014, Bogaard and Greco, 2015, and references therein). On the other hand, thinking at the infinite slope factor of safety FS formula (see again references above) the relationship between piezometric depth and displacement may be expected to be linear (this however assumes a linear relationship between FS and displacement). So what I expect is: non-linear correlation for rainfall-piezometric depth, linear for piezometric depth-displacement. In fact, this is reflected on the values of max CCF reported in tab 1 and tab 3 respectively. The authors should make scatter plots of one variable against the other at various time lags, in order to assess whether or not the statistical dependence between variables is LINEAR or NON-LINEAR. In the second case the authors should apply: or a transformation of the variables, or a non-linear (cross) correlation analysis. This is a crucial issue that the authors need to address for publishing the paper

Fig. 2: Significance of statistical analysis may be improved by adding other data. The data presented in the paper cover the years 1999-2002. From papers by the same authors it seems that other data do exists (e.g. Corominas et al, 2005). If this is the case, why not add these data to the analysis?

Sect. 5: Paper organization mistake in the Results section: subsections report results relative to a variables combination that is different from that declared in the subsection title. The discussion of rainfall vs piezometric depth is missing. In detail: title of 5.1. should be rainfall vs displacement, 5.2 piezometric depth vs displacement, 5.3. displacement vs displacement, 5.4. piezometric depth vs piezometric depth. 5.5 is a repetition of 5.4

Discussion (Section 6): Some of the conclusions seem to be not directly supported by the paper results, and are a rather subjective interpretation of the authors. Please better link discussion to results, and explicitly declare what should be assumed as a subjective/reasonable interpretation of the authors

[Figure]

P8 L16-19: please explain better the "second mechanism"

Fig. 2b: is the plot of piezometric depth for S4 correct?

Technical corrections

P2 L6 perhaps replace "monitoring data" with "landslide-related variables" P2 L11: first 950 m then 1250 m; it should also be specified that elevation is measured a.s.l. (above sea level)

Is there a specific reason why borehole numbering has to be S4, S2 and S9? If not, why not renumber as SL1 SL2 and SL3 (where L indicates "Lower Unit")?

Tables 1-5 replace "lap" with "lag"

P10 L13 remove "350"

References (of review):

Baum, R. L., Godt, J. W., and Savage, W. Z.: Estimating the timing and location of shallow rainfall-induced landslides using a model for transient, unsaturated infiltration, J. Geophys. Res., 115, F03013, doi:10.1029/2009JF001321, 2010.

Bogaard, TA, R Greco (2015) Landslide hydrology: from hydrology to pore pressure. WIREs Water 2015. doi: 10.1002/wat2.1126

D'Odorico, P., Fagherazzi, S., and Rigon, R.: Potential for landsliding: Dependence on hyetograph characteristics, J. Geophys. Res.-Earth Surf., 110, F01007, doi:10.1029/2004JF000127, 2005.

Iverson, R. M.: Landslide triggering by rain infiltration, Water Resour. Res., 36, 1897–1910, 2000.

Peres, D. J. and Cancelliere, A.: Derivation and evaluation of landslide-triggering thresholds by a Monte Carlo approach, Hydrol. Earth Syst. Sci., 18, 4913-4931, doi:10.5194/hess-18-4913-2014, 2014.

Salas, J. D.: Analysis and Modeling of Hydrologic Time Series, Chap. 19, Handbook of Hydrology, McGraw-Hill, 1993.

---

## Referee Comment (RC2) · Anonymous Referee #2 · 23 Nov 2016

The paper presents a statistical analysis of monitoring observations at the Vallcebre landslide for a time period not already published in (the many) other manuscripts documenting the behaviour of this landslide.

The novelty of the approach and findings is not significant for several reasons: - linear correlation is applied which is extremely questionnable for geological processes highly influenced by non-linear relationships and transients? - the authors filtered the data applying signal-to-noise methods to remove instrumental errors – what is the effect of this SNR ? how many data were removed and what is the influence on the correlations?

[Figure]

- seasonal patterns are observed in the time series, and seasonal detrending should be applied before applying CCF – it is not very clear how the authors pre-processed their data. - the observation time series are also questionnable. For instance, it has been demonstrated by several authors that effective rainfall is better correlated to piezometric variations, than net cumulative rainfall. What is the argument of using net rainfall for the analysis? Further, snow might impact the water budget. Did the authors consider th possible additional input of waters on the slope? Further, the piezometric depths should be transformed in hydraulic heads or better in pore pressures above the slip surface for a consistent analysis.

Further, the authors should discuss the characteristics of the studied period regarding the long-term evolution of the slope. Is the period 1999-2001 representative of a low/high geomorphological activity of the slope, or a period of interest because many data/sensors were available? Some justification is needed especially because the approach could be tested on the complete monitoring dataset available for the landslide (at least for some combinations of parameters such as rain and displacement). This would possibly give more significance to the work and reveal some changes in the behaviour in time.

The discussion section is weak. I would like to have a discussion on the signifiance of the time lag statistically calculated by the authors with regard to the many hydrological models that were applied on this slope.

I conclude that the manuscript has to be rejected for NHESS.
* * *

---

## Author Comment (AC1) · 3 Jan 2017

**Answers to Comments from Anonymous Reviewer #1**

R1_Comment #1:

"…Literature cited in the paper may be enriched (for instance, for the early warning and the modeling part – first paragraph of introduction, P1 L15-19)…"

R1_Answer #1:

Thanks for the suggestion we will extend bibliographic reference.

R1_Comment #2:

"…P1 L20: Cross-correlation is a quite standard statistical tool (see Handbook of hydrology, Salas, 1993 - this must be cited)…"

R1_Answer #2:

Thanks for the suggestion, we will integrate this bibliographic reference

R1_Comment #3:

"…P4 L20: Please provide more details on how the confidence lines have been determined (statistical significance threshold), i.e. the formula used…"

R1_Answer #3:

Confidence interval are the same as used in medical literature (*i.e.* Chaves, L.F., Pascual, M., 2006. Climate Cycles and Forecasts of Cutaneous Leishmaniasis, a Nonstationary Vector-Borne Disease. PLOS Med. 3, 1–9. doi:10.1371/journal.pmed.0030295). More in detail, the blue dashed lines indicate the 95% confidence intervals for the cross-correlation between two time-series composed by "*white noise*" with the number of samples ("n") equal to samples composing the monitoring time-series (Brockwell PJ, Davis RA (2002) Introduction to time series and forecasting, 2nd ed. New York: Springer. 434 p.)

$$CCF\ 95\% = \frac{2}{\sqrt[2]{n}}$$

Where "*n*" is the number of samples forming the time-series.

We recall, as explained in the manuscript P4 L20, that cross-correlation results have been considered acceptable if they were either above (if positive) or below (if negative) the +/- confidence threshold. Additionally, since some of the cross-correlation maxima were quite low, we have now computed the p-value corresponding to the identified cross-correlation maxima in order to strengthen our assessment. More in detail, by means of the p-value, the null hypothesis tested corresponds to the absence of correlation between variables. The results show p-values close to zero (from 0 to $7.8 \times 10^{-98}$) for all combinations, which support the significance of cross correlation results even in the cases of low maxima.

**R1_Comment #4:**

"…P4L23 Why a 5% threshold has been chosen, and not another one? The authors should possibly investigate if statistical tests aimed at verifying the significance of cross-correlation variation do exist. Though this issue may significantly change the time-lag interval around the maximum CCF value, the 5% threshold may be however accepted. What I just ask to the authors is to possibly justify this value and to verify the existence of above-mentioned statistical tests…"

**R1_Answer #4:**

Since most correlograms show CCF distributions that are somehow similar to "Gaussian", we decided to include the values of CCF at +/- 5% threshold from the peak (and the related time-lag intervals) in order to provide information about the "shape" (flatness: i.e. lower significance" or peakness, i.e. higher significance) of the cross-correlogram without having to provide figures for all cross-correlogram. And yes, we might have used a different value as well. The choice of 5% was arbitrary, and driven by the fact that generally, this is a familiar value in Gaussian distributions. However, since the correlograms are not precisely Gaussian, we agree with the reviewer that the 5% might not represent the same statistical significance for all correlograms (i.e. we might have asymmetric distribution around the peak), and this might be a weakness. Nevertheless, we still believe that in order to "intuitively" compare the different correlograms that cannot be reduced to a precise Gaussian distribution, it is necessary to use the same +/- threshold, even if it might not necessarily represent the same level of significance of cross-correlation. Thus, we believe that for the purposes it was meant for, the analysis of the statistical significance of the adopted +/- range would not add much information about the reliability of the correlation peak in terms of time-lag interval.

**R1_Comment #5:**

"…The authors should make scatter plots of one variable against the other at various time lags, in order to assess whether or not the statistical dependence between variables is LINEAR or NON-LINEAR. In the second case the authors should apply: or a transformation of the variables, or a non-linear (cross) correlation analysis. This is a crucial issue that the authors need to address for publishing the paper."

**R1_Answer #5:**

We have performed the requested time-lagged scattered plots and, as correctly assessed by the reviewer, non-linearity affects relationships "rainfall vs displacements" and "rainfall vs. piezometric depth". At the same time we can confirm that a linear dependency can be reasonably assumed to exist ($R^2$ values between 0.6 and 0.78) for the following time series combinations: Piez. Depth vs. Piez. Depth.; Displ. Rate Vs- Displ. Rate; Piez.depth vs. Displacement Rate. However, it is also to be mentioned that, despite the fact that Corominas et al (2005) indicated non linearity between Piez. depth vs. Displacement in the long term, the scattered plots indicate linearity over the short term. Moreover, it should be noticed that we've here more explicitly indicated that, in order to apply the cross-correlation function (see also response to R2_Comment#1c), the displacement time-series had already been, in the submitted paper, converted into differential displacements, i.e. the displacement occurred within the 20 minutes sampling interval, which is essentially a displacement rate (velocity). Finally, we must say that, unfortunately, the non-linear correlation between precipitation and other variables cannot be reduced to any power law, so transformation (i.e. linearization) is basically impossible. Thus, being fully aware that this is the crucial issue for the acceptability of the paper, we propose to the Editor that we make a major revision of the paper by :

- highlight the non-linearity between rainfall vs. piezometric level" and "rainfall vs. displacements" and discuss their dependence on a more qualitative level
- highlight, in order to avoid misunderstandings, that CCF was applied to "displacement rate" and not, as reader might erroneously assume, cumulative displacements
- limit the CCF analysis to the combinations that proved to have an acceptable linear dependency: we believe we would still be able to discuss the Vallcebre landslide dynamics under several perspectives, such as hydrogeological features (by means of "Piez. Depth vs. Piez. Depth" cross-correlation results), style of movement (by interpreting results from the cross-correlation between "Displ. RATE Vs- Displ RATE" time-series), hydro-mechanical processes (highlighted by the results of cross-correlation between "Piezometric depth vs. Displacement *RATE*").
- add an appendix section with the lagged scatterplot (with $R^2$ values respect to linear trends and specific p-values) will be included in order to present the degree of linear dependency between analysed variables by means of the cross-correlation function.
* * *
R1_Comment #6:
"…Fig. 2: Significance of statistical analysis may be improved by adding other data. The data presented in the paper cover the years 1999-2002. From papers by the same authors it seems that other data do exists (e.g. Corominas et al, 2005). If this is the case, why not add these data to the analysis?

R1_Answer #6:
Yes, other data exist. However, we can argue that: (i) the analysed time interval (from 01-Jan-1999 to 01-jan-2002) covers 3 years characterized by variations of velocity and it is in any case

representative of the "ordinary" mobilization pattern of the landslide (ii) the analyzed time interval it is the longest available interval characterized by full continuity of data. So yes, we might have analyzed also other periods, but on separate calculations, since continuity of the time-series is a discriminant for the application of the cross-correlation function. To exemplify our arguments, we might include in the revised paper (if the editor believes it might be necessary) the figure 1, which shows the average displacement trend of the landslide ($\cong$25cm/year) over the 15 years-period of measurements, evidencing how the analyzed period is in line with all other "ordinary" years (that are different from the "unusual" period 1997-1998, which was characterized by velocities higher than the usual ($\cong$50 cm/year)), and how, after 2002, some gaps start to appear in the time series.

[Figure]

*Figure 1: Cumulative displacements of the wire extensometer S-2 during the period 1996-2012.*
* * *
**R1_Comment #7:**

Sect. 5: Paper organization mistake in the Results section: subsections report results relative to a variables combination that is different from that declared in the subsection title. The discussion of rainfall vs piezometric depth is missing. In detail: title of 5.1. should be rainfall vs displacement, 5.2 piezometric depth vs displacement, 5.3. displacement vs displacement, 5.4. piezometric depth vs piezometric depth. 5.5 is a repetition of 5.4.

**R1_Answer #7:**

Thanks, corrections will be performed.
* * *
**R1_Comment #8:**

"…Discussion (Section 6): Some of the conclusions seem to be not directly supported by the paper results, and are a rather subjective interpretation of the authors. Please better link discussion to results, and explicitly declare what should be assumed as a subjective/reasonable interpretation of the authors…"

**R1_Answer #8:**

We can edit the discussion, so to meet the reviewer requirements, in the following way:

The results obtained by means of CCF analysis can be discussed on the perspective of hydro-mechanical slope processes that are somehow more complicated that a direct relationship between groundwater and movements.

The presence of lagged response in open-pipe piezometers like those installed in the Vallcebre landslide should be beard in mind during the discussion of the results. Depending on the piezometer type, geometry and on the permeability of the local soil around it, different time lags are necessary to measure the 90% of the occurred piezometric level variation. Nevertheless, it should be taken in consideration that a complete level variation is not necessary for the CCF analysis to determine a time-lag value.

The time-lags between different piezometers from upslope to downslope range predominantly in the positive values (Fig. 7f), meaning that the response of each piezometer to groundwater level is driven by the variations occurred upslope. It is therefore reasonable to consider as a working hypothesis the presence of a downslope directed pore-pressure transfer-wave that, even in substantial absence of water transfer (given the low permeability of landslide materials), determines the major peaks of groundwater levels after rainfall. Therefore, pore-pressure transfer results to be the hydrological key-factor determining groundwater level variations in this portion of the landslide body rather than an actual groundwater filtration involving mass transfer. A tentative estimation of the apparent pore-pressure velocity transfer can be done by considering the time-lags obtained in the cross-correlation between piezometric depths and the distance between piezometers. On a such basis, the apparent pore-pressure velocity transfer can be estimated as: $2.1×10-2$ m s-1 from S4 to S2; $4.1×10-1$ m s-1 from S2 to S9; $3.1×10-1$ m s-1 from S4 to S9. It can be speculated that the higher apparent velocity obtained in the sector from S2 to S9, in the lowest part of the landslide unit, might be related to the fact that it corresponds to a compression zone where, presumably, the pore-pressure transfer through the landslide body is more effective.

On a mechanical perspective, the cross correlation between displacements recorded by the wire-extensometers indicates that it is quite reliable to consider the assessment of a retrogressive evolution of movement propagation, with time series of displacement in downslope extensometers leading time series of displacement in upslope extensometers (see Fig. 7e for a synoptic view). The aforementioned evidence is in agreement with the results of the landslide evolution model proposed by Ferrari et al. (2011), in which the key-role of the toe erosion by the Vallcebre stream for the triggering of the landslide is stressed. On a hydro-mechanical perspective, the analysis of the relationship between piezometric depth and displacements confirm the leading role of piezometric depth variation since in all three monitoring sites displacement peaks result to occur after ground water peaks (Fig. 7b-c).

Considering all the aforementioned findings, it is therefore reasonable to assume that the response of the Vallcebre landslide is led by the following mechanism: the increase of pore water pressures is first noticed in the upper part of the landslide unit (S4 and S2), it is not high enough to produce the acceleration of the landslide. Once the increase of pore water pressure reaches the landslide foot (S9), then the acceleration takes place. Therefore, motion is driven from the foot of the landslide to its head. Thereafter, the advance of the landslide mass increase instability of the landslide foot, which combined with the torrent erosion, cause periodic small failures there, delivering slope

material to the valley bottom subsequently evacuated by the stream (Fig. 8). This triggers a retrogressive evolution of slope movements that affects all the lower unit of the Vallcebre landslide.
* * *
R1_Comment #9:

"…Sect P8 L16-19: please explain better the "second mechanism"…"

R1_Answer #9:

In the previous point we propose to edit the discussion, so to meet the reviewer requirements. Last paragraph essentially explains the mobilization mechanism occurring in the landslide (see answer above).

Eventually, the following sentence and the figure 2 can be added: *"We have evidences that the slope toe is being eroded (see Figure 2) and this erosion may lead to further displacement of the landslide foot as shown by Ferrari et al (2008). The interaction between the toe of the landslide and the Vallcebre torrent can be noticed in figure 2, were the evolution in time of the distance of the to respect to a benchmarck (boulder in the stream) can be followed through ten years (from 2003 to 2013). However, to erode the foot a minimum discharge of the Vallcebre torrent is required.*

*2003*

[Figure]

*2008*

[Figure]

*2013*

[Figure]

Figure 2 (top): local failure of slope toe deposited in the torrent bed; the deposits are removed by erosive activity of the torrent (middle) and new local failures are generated (bottom).
* * *
R1_Comment #10:
"…Sect Fig. 2b: is the plot of piezometric depth for S4 correct?..."
R1_Answer #10:
Yes, this is the groundwater depth recorded at the S4 site

R1_Comment #11:
"…P2 L6 perhaps replace "monitoring data" with "landslide-related variables" P2 L11: first 950 m then 1250 m; it should also be specified that elevation is measured a.s.l. (above sea level)…"
R1_Answer #11:
Thanks, we will follow your indications.

R1_Comment #12:
*Is there a specific reason why borehole numbering has to be S4, S2 and S9? If not, why not renumber as SL1 SL2 and SL3 (where L indicates "Lower Unit")?*
R1_Answer #12:
We have kept the borehole numbering presented in the previous papers (i.e. Corominas et al. 2005) in order to maintain consistency.

R1_Comment #13:
*Sect Tables 1-5 replace "lap" with "lag"*
R1_Answer #13:
Thank, the correction will be performed.

R1_Comment #14:
*P10 L13 remove "350"*
R1_Answer #14:
Thanks, we will erase it.

---

## Author Comment (AC2) · 3 Jan 2017

**Answers to Comments from Anonymous Reviewer #2:**
* * *
**R2_Comment #1a:**

"…The novelty of the approach and findings is not significant for several reasons: - linear correlation is applied which is extremely questionable for geological processes highly influenced by non-linear relationships and transients?..."

**R2_Answer #1a:**

The R1_Answer#5 given to Reviewer 1 can also apply to this comment. In summary, we have performed time-lagged scattered plots tests to investigate the linear relationship between variables and, actually, we found out that a linear dependency can be assumed to exist ($R^2$ values between 0.6 and 0.78) for the following time series combinations: Piez. Depth vs. Piez. Depth.; Displ. RATE Vs- Displ. RATE; Piez.depth vs. Displacement RATE. Thus, we believe that the cross correlation analysis of such variables is fully justified and, at the same time (as specified in R1_Answer #5) we propose to eliminate the cross-correlations that have proven non-linear, which are only these including rainfall.
* * *
**R2_Comment #1b:**

"…- the authors filtered the data applying signal-to-noise methods to remove instrumental errors – what is the effect of this SNR ? how many data were removed and what is the influence on the correlations?..."

**R2_Answer #1b:**

As presented in the paper, we have applied a Fourier Filtering that operates in the frequency domain. This have been done after a careful observation of the raw time-series evidencing "noise" that appeared after the data-logger were replaced (such replacement was made prior to the  time period analyzed in this paper). As a consequence, after having identified the exact temporal frequency of this electric noise (which corresponds to a return period of 30h), we have selectively removed it. Therefore, filtering has not influenced the cross correlation of our time-series, as it is clearly visible in figure 5, were both the raw and the filtered time-series are presented.
* * *
**R2_Comment #1c:**

"… seasonal patterns are observed in the time series, and seasonal de-trending should be applied before applying CCF – it is not very clear how the authors pre-processed their data…"

**R2_Answer #1c:**

To apply the cross-correlation function the displacement time-series have been converted into differential displacements, i.e. the displacement occurred within the 20 minutes sampling interval, which is essentially a displacement rate (velocity). This allowed avoiding any complicated de-trending processing. In order to avoid misunderstandings, we'll therefore change any reference to displacements into "displacement rate". For the same reason, displacement rate will be plotted in figure 2a instead of cumulative displacements.

**R2_Comment #1d:**

"…- the observation time series are also questionable…"

**R2_Answer #1d:**

The observation time-series represent a rare case of long (three years) monitoring series recorded at a landslide site. Previous papers on the Vallcebre landslide presenting the monitoring devices used represent a benchmark for the Engineering Geology scientific community. Furthermore, the high frequency sampling of 20 minutes gives space to very accurate considerations of the occurring geological processes. So we fail to see the justification of this comment.

**R2_Comment #2:**

"…For instance, it has been demonstrated by several authors that effective rainfall is better correlated to piezometric variations, than net cumulative rainfall. What is the argument of using net rainfall for the analysis?…"

**R2_Answer #2:**

Since the analysis proposed is not focused in the definition of the daily relationship within the variables analyzed but in evidencing the global dependencies resulting in a long term perspective, we decided to use the net rainfall. As a matter of fact, the use of net rainfall would result equilibrated at the end of each hydrologic year. In any case, because of previous comments pointing to non-linear relation, that has been verified, we propose to eliminate the rainfall parameter from the cross-correlation analysis so that R2_Comment #3 is implicitly satisfied.

**R2_Comment #3:**

"…Further, snow might impact the water budget. Did the authors consider the possible additional input of waters on the slope?…"

**R2_Answer #3:**

Since we plan to revise the paper eliminating cross-correlation regarding precipitation, (rains and snowfalls) the R2_Comment #3 will be implicitly satisfied. Nevertheless, it is worthwhile recalling that, unfortunately, we have no reliable statistics on the snow fall in Vallcebre since there are no direct measures of it. In the past, Viladrich L (1989 - Neva o no neva? . Erol, 26: 41-45 (in catalan)) recorded an average of 9 snow events per year for the period 1958-1988, mostly concentrated between December and April. In his report, a snow event may consist of few snowflakes melting after the contact with the ground and snow fall followed by rain. For our experience, snow stands occasionally on the ground at Vallcebre. Most of the fallen snow melts during the next few days. However, during some big events, such as in December 1996, January 1997, or in March 2010 (outside the analyzed period), the snow may have lasted on the ground for more than one week. As mentioned in the manuscript (p.4 lines 5-6), all the wire extensometers display a seasonal trend, with accelerations in spring and fall (periods with higher

rainfall rate) and a number of short term acceleration periods after specific precipitation events. As there is no permanent snow cover in winter, we believe that it is reasonable to assume that the acceleration is mostly controlled by the rainfall pattern and, to a lesser extent, by the toe erosion (this is an assumption that has been confirmed by the results of the present manuscript).

**R2_Comment #4:**
"…Further, the piezometric depths should be transformed in hydraulic heads or better in pore pressures above the slip surface for a consistent analysis…"

**R2_Answer #4:**
We do not believe that this is necessary since we are not performing limit equilibrium analysis or physical modelling. The time series of pore pressure would have the same pattern in time of the piezometric depth time-series. It is noteworthy to recall that in cross-correlation analysis what matters is the pattern of the time series, i.e. variation in time, of the processed time-series. Therefore, transforming piezometric depth into hydraulic heads or *pore pressures above the slip surface* would not give any additional information to the analysis.

**R2_Comment #5:**
"…Further, the authors should discuss the characteristics of the studied period regarding the long-term evolution of the slope. Is the period 1999-2001 representative of a low/high geomorphological activity of the slope, or a period of interest because many data/sensors were available? Some justification is needed especially because the approach could be tested on the complete monitoring dataset available for the landslide (at least for some combinations of parameters such as rain and displacement). This would possibly give more significance to the work and reveal some changes in the behavior in time…"

**R2_Answer #5:**
We provide here the same answer given to reviewer 1 (see "R1_Answer #6").

Yes, other data exist. However, we can argue that: (i) the analyzed time interval (from 01-Jan-1999 to 01-jan-2002) covers 3 years characterized by variations of velocity and it is in any case representative of the "ordinary" mobilization pattern of the landslide (ii) the analyzed time interval it is the longest available interval characterized by full continuity of data. So yes, we might have analyzed also other periods, but on separate calculations, since continuity of the time-series is a discriminant for the application of the cross-correlation function. To exemplify our arguments, we might include in the revised paper (if the editor believes it might be necessary) the figure 1, which shows the average displacement trend of the landslide ($\cong$25cm/year) over the 15 years-period of measurements , evidencing  how the analyzed period is in line with all other "ordinary" years (that are different from the "unusual" period 1997-1998, which was characterized by velocities higher than the usual ($\cong$50 cm/year)), and how, after 2002, some gaps start to appear in the time series.

[Figure]

*Figure 1: Cumulative displacements of the wire extensometer S-2 during the period 1996-2012.*

**R2_Comment #6:**

"…The discussion section is weak. I would like to have a discussion on the significance of the time lag statistically calculated by the authors with regard to the many hydrological models that were applied on this slope…"

**R2_Answer #6:**

Actually, no complete hydrological analysis/modelling has been carried out so far. We run a hydrological model (Transin) to calibrate the hydraulic parameters modelled (permeability and heads) against the values obtained with the pumping tests and the observed groundwater table (Corominas et al 2008). In the analysis performed in Corominas et al. (2005) we computed the landslide displacements and velocities from groundwater level changes considering a viscous term. In the latter work we mentioned (p. 90) that peak water levels were attained at different times, depending on the permeability of the adjacent material. Boreholes (S4) located on the graben had a faster response and drainage than the rest. We also observed (p. 91) some synchronism between the groundwater level changes and the displacements at S2 and a lack of correlation of the event in January 1997, which could be caused by toe erosion by the Vallcebre torrent. This was only an hypothesis that this work with CCF supports.